# Brief communication: Increasing shortwave absorption over the Arctic Ocean is not balanced by trends in the Antarctic

Christian Katlein[1], Stefan Hendricks[1], Jeffrey Key[2]

[1]Alfred-Wegener-Institut Helmholtz-Zentrum für Polar- und Meeresforschung, Bremerhaven, 27570, Germany
[2]Center for Satellite Applications and Research, NOAA/NESDIS, Madison, Wisconsin, USA

*Correspondence to*: Christian Katlein (Christian.Katlein@awi.de)

**Abstract** On the basis of a new, consistent, long-term observational satellite dataset we show that despite the observed increase of sea ice extent in the Antarctic, absorption of solar shortwave radiation in the southern ocean poleward of 60° latitude is not decreasing. The observations hence show that the small increase in Antarctic sea ice extent does not

compensate for the combined effect of retreating Arctic sea ice and changes in cloud cover, which both result in a total increase in solar shortwave energy deposited into the polar oceans.

## 1 Introduction

Changes in the Arctic and Antarctic cryosphere have been continuously monitored by different satellite programs since the

1970's. Arctic sea ice is becoming thinner (Haas, Pfaffling et al. 2008) and younger (Maslanik, Fowler et al. 2007) coupled with a decline in its extent (Serreze, Holland et al. 2007, Stroeve, Serreze et al. 2012). This leads to a decrease in area-average sunlight reflection, and thus to higher energy absorption in the Arctic Ocean (Perovich, Jones et al. 2011, Nicolaus, Katlein et al. 2012). While some areas in Antarctica have also experienced a reduction of the sea ice cover, a modest overall gain of sea ice area has been observed in the Southern Hemisphere (Cavalieri, Gloersen et al. 1997, Stammerjohn, Massom

et al. 2012) though there is some uncertainty in the magnitude of the trend (Eisenman, Meier et al. 2014). How these opposing trends relate to each other on a global scale is governed by a multitude of factors, such as the different latitudinal position of the ice cover and constraints by land masses, significant differences in the physical properties of the ice surface, and different forcing mechanisms from lower latitudes (Meehl, Arblaster et al. 2016). A particular problem in relating Arctic and Antarctic trends and understanding their global impacts is the lack of a long-term consistent observational dataset

covering both poles.

The increased absorption of sunlight due to the loss of sea ice results in ocean warming, more ice loss, a decrease in albedo, and a further increase in absorbed sunlight. This is known as the ice-albedo feedback (Curry, Schramm et al. 1995), a critical process in the global shortwave energy budget. Most of this added heat will be lost due to increased longwave emission during winter and will not be carried on into the next year. Especially in the Arctic, a longer sea ice melt season

(Markus, Stroeve et al. 2009), thinner ice (Haas, Pfaffling et al. 2008), and increased melt-pond coverage (Rösel and Kaleschke 2012) lead to increasing solar shortwave energy deposition in the ice-ocean system (Nicolaus, Katlein et al. 2012), adding to the increase in absorption due to decreasing sea ice extent (Pistone, Eisenman et al. 2014). However, surface characteristics of Antarctic sea ice are less affected by global climate change (Allison, Brandt et al. 1993, Brandt, Warren et al. 2005, Laine 2008). Antarctic sea ice is mainly melting from below as the ice drifts away from the continent into warmer circumpolar waters, in contrast to the surface melting induced by melt ponding on Artic sea ice. Therefore, sea-ice extent losses in the Arctic are most pronounced during the Northern Hemisphere summer. In the Antarctic, the increasing extent of Antarctic sea ice is observed during the Southern Hemisphere winter, when the impact of sea ice cover on the shortwave energy balance is weaker.

Here we evaluate observations of the combined effect of different radiative processes in both hemispheres on the shortwave energy budget of the polar oceans. Our goal is to determine to what extent the increased absorption of solar shortwave energy caused by losses and other changes in Arctic summer sea ice are offset by potentially decreased absorption due to observed increases of sea ice extent in the Antarctic. The recently published Advanced Very High Resolution (AVHRR) Polar Pathfinder - Extended (APP-x) dataset provides a novel tool to investigate this question on the global scale. It provides surface radiative properties and fluxes consistently derived twice daily (high and low sun) for both polar regions beginning in 1982 (Key, Wang et al. 2016). Its great advantage compared to earlier, individual products based on AVHRR data is that the integrated dataset inherently takes into account changes in cloud cover and albedo changes of various sources, allowing us to evaluate the actual shortwave energy deposition changes in the oceans poleward of 60° latitude. We calculated monthly averaged shortwave radiative fluxes into the ice-ocean system to estimate the partitioning of absorbed shortwave energy between sea ice and the unfrozen ocean surface.

**Methods**

The results presented here are based on version 1.0 of the Extended AVHRR (Advanced Very High Resolution Radiometer) Polar Pathfinder (APP-x) data. APP-x contains twice-daily data of many surface, cloud, and radiative properties retrieved at high sun and low sun times (04:00 and 14:00 local solar time for the Arctic; 02:00 and 14:00 for the Antarctic) from satellite data using a suite of algorithms and a radiative transfer model (Key, Wang et al. 2016). All variables are derived from the same set of satellite radiances, allowing an integrated view on the effects of sea-ice changes. We use the variables ice thickness, surface albedo, cloud cover and downwelling shortwave radiation at the surface. The APP-x record begins in 1982 and continues to the present day, though version 1.0 used in this study covers the period 1982-2014. Through validation studies, the APP-x variables used here have been determined to be of sufficient accuracy to be considered as climate data record variables. APP-x shortwave fluxes have been validated against observations from the SHEBA ice camp and the CERES satellite product (Riihelä, Key et al. 2017). Details on the retrieval of the variables can be found in the respective Climate                Algorithm                Theoretical                Basis                Document

 APP-x utilizes the Near-real-time Ice and Snow Extent (NISE) product from the National Snow and Ice Data Center (Boulder, Colorado, USA) for surface type identification.

The energy flux absorbed by the surface was calculated as

$$E = E_{down} \cdot (1 - \alpha) \,, \tag{1}$$

and multiplied by 12 hours and the grid cell size to obtain the total amount of absorbed energy, where $E_{down}$ inherently
accounts for cloud cover. All grid cells with ice thickness greater than 0 were considered as ice covered. The APP-x data product does not contain a separate field for ice-concentration, but ice thickness is only calculated for cells with an ice concentration >15% in the NISE product. Sea ice extent was calculated as the number of ice-covered grid cells multiplied by the cell size (25x25 km). This yields slightly larger numbers than comparable analyses on the direct basis of passive microwave sea-ice concentration products with higher resolution, but the magnitude of changes proved to be unaffected.
Shortwave energy fluxes were calculated for the twice-daily data and averaged over each month to reduce the influence of retrieval errors and intermittent gaps in the data. For the calculation of total absorbed energy, twice daily data was summed up to monthly values. For grid cells with invalid retrievals due to low light during winter, shortwave fluxes were set to zero and the albedo was set to one. Monthly data of average energy flux and total absorbed energy were then used for annual and long-term averages as well as for time series analysis. Data for the year 1994 were excluded from time
series analysis due to a significant gap in the observations. Trends were calculated through linear regression using the Matlab curve-fitting toolbox. All trends presented as significant have confidence levels above 95%. Given error intervals for trends are 95% confidence intervals.

**Results**

An analysis of the dataset revealed a decrease of September sea-ice extent of -0.126 ($\pm$ 0.03) million km²/year for the
Northern Hemisphere and an insignificant increase of 0.012 ($\pm$ 0.02) million km²/year in the Southern Hemisphere summer in March (Figure 1a). Antarctic sea-ice extent increased 0.020 ($\pm$ 0.01) million km²/year in September during Southern Hemisphere winter, while winter sea ice loss in the Arctic is weaker in March with a loss of -0.041 ($\pm$ 0.01) million km²/year. Arctic sea-ice extent losses are thus five times larger in magnitude than the small increases in ice area in the Antarctic, leading to a combined total sea ice loss of -0.106 million ($\pm$ 0.03) km²/year in September and -0.028 ($\pm$ 0.025)
million km²/year in March. This reproduces the known global net loss of sea-ice covered area during the last few decades (Stammerjohn and Smith 1997, Stammerjohn, Massom et al. 2012, Stroeve, Serreze et al. 2012). Thus, the slight gains in Antarctic sea-ice area do not compensate for the areal loss of sea ice in the Arctic. Given the differences in the latitudinal

distribution of Arctic and Antarctic sea ice, however, the impact of these changes on the absorption of solar radiation warrants further investigation.

The summer mean daytime albedo for ice-covered areas in the APP-x dataset was 0.34 for the Arctic (June/July/August) and 0.41 for the Antarctic (December/January/February) which compares well to earlier studies (Allison, Brandt et al. 1993). The higher albedo of Antarctic sea ice may be caused mainly by a thicker snow cover with little surface melt and consequently the lack of melt water pond formation. In accordance with the observed trend towards younger, predominantly seasonal Arctic sea ice with larger melt pond coverage, the APP-x dataset shows a decrease of the mean Arctic summer sea ice albedo, while in the Antarctic albedo trends show regional differences driven by the changes in ice concentration (Figure 2). In this analysis of albedo trends, we only consider summer daytime albedos, as the albedos retrieved during wintertime are questionable due to low light levels and observation gaps. This does not significantly affect our analysis of energy fluxes, as the largest uncertainty in the albedo occurs with low fluxes subsequently leading to a low energy flux uncertainty.

Antarctic sea ice exists mainly in the latitude zone between 55 and 77°S. In contrast, Arctic sea ice occupies the region between approximately 70 and 90°N. Due to its generally higher snow cover (Massom, Eicken et al. 2001) and the 20% higher albedo as well as its location at lower latitudes with higher shortwave insolation, the presence of sea ice does have a stronger impact on the local shortwave energy balance in the Antarctic. Mean annual shortwave energy uptake by the ice-ocean system polewards of 60° latitude was calculated twice daily from APP-x surface albedo and incoming solar radiation at the surface and averaged for each month. The use of these APP-x quantities inherently accounts for trends in cloud cover and surface albedo changes.

The mean annual shortwave energy flux into the ice-ocean system poleward of 60° accounts for 68 $W/m^2$ in the Arctic, and 60 W/m² in the Antarctic (Figure 1b). Average Southern Hemisphere absorption shows high interannual variability throughout the satellite record, while the absorbed flux of the ice-ocean system in the Arctic clearly increased by 0.48 $W/m^2$ per year. While the trend in the absorption of solar shortwave energy is more uniform across the Arctic Ocean due to uniform sea ice changes, there are large regional differences in the Antarctic (Figure 2c,d). More solar energy is absorbed in the Bellingshausen and Amundsen Seas, with smaller areas of decreasing energy absorption.

Combining the effects of cloud cover induced insolation changes, reduced sea ice extent and a lower surface albedo, the total annual shortwave energy absorbed by the ice-ocean system north of 60°N increased at a rate of $1.7 \, (\pm 0.5) \cdot 10^{20}$ J/yr . In the Southern Hemisphere energy absorption by the ice-ocean system south of 60°S also increased at a very similar rate of $1.8 \, (\pm 1.9) \cdot 10^{20}$ J/yr but due to the large interannual variability the trend is not statistically significant. Despite the increasing winter sea ice extent in the south, both hemispheres show a distinct increase in energy deposition in the ice-ocean system leading to ice melt and ocean warming. Increased ice extent in the Antarctic therefore does not decrease annual mean energy absorption as might be expected. An analysis of anomalies in sea ice extent, albedo and shortwave energy deposition in the ice-ocean system shows that energy deposition is not directly correlated with the ice extent and

albedo anomalies in general but can be offset by changes in cloud cover leading to increasing shortwave energy absorption in spite of albedo increases (Figure 3).

In the Antarctic, the energy flux anomaly does not exhibit a strong relationship with ice extent anomalies overall (Figure 3b,d), while in the Arctic anomalies in albedo and the resulting heat input into the ice-ocean system are much more closely related to the sea ice extent anomaly (Figure 3a,c,e). Thus, in the context of the surface shortwave radiation balance, losses in Arctic sea ice and the resulting increase in solar energy absorption are not balanced by the moderate gains in sea ice extent in the Antarctic. On the global scale, changes in the shortwave energy partitioning in the polar oceans poleward of 60° latitude lead to a combined increased energy deposition of $3.5(\pm 2.1) \cdot 10^{20}$ J/yr comprised of positive energy absorption trends in both hemispheres despite moderate increases in Antarctic sea ice extent.

When extending our analysis from the oceans to all land and ocean areas poleward of 60° latitude, the result is similar with an increasing flux of 0.3 W/m$^2$ per year absorbed by the planet's surface poleward of 60°. This trend is somewhat weaker than over the ocean alone, as changes in land cover properties are not as pronounced as changes in ice extent and properties. Still, changing snow cover and prolonged melting seasons cause more absorption of sunlight and further heating of the climate system.

**Conclusion**

In conclusion, a consistent long-term observational, satellite-based time series shows that changes in the shortwave energy budget caused by a decreasing Arctic sea ice cover are not balanced by the slight increases observed in Antarctic sea ice extent. Significant increases in Antarctic sea ice only occur during the Southern Hemisphere winter and thus have only a minor impact on the energy balance, while Arctic sea ice changes are accompanied by a spatially uniform decrease of sea-ice albedo during the summer, further increasing the energy input to the northern polar ocean and thereby strengthening the ice-albedo feedback. This demonstrates the different roles that partitioning of solar shortwave radiation plays in the two very different sea ice zones of our planet. It is necessary to better understand the influence of the changes in different processes such as sea ice distribution, sea ice albedo and cloud properties on the shortwave radiation budget in these two very different polar regions.

**Author Contributions**

CK performed the calculations, prepared the figures and wrote the text, SH had the initial idea for this study and contributed to the setup of the study, JK contributed the APP-x data and provided insight into its use. All authors were involved in the writing of the manuscript. The authors declare that they have no conflict of interest.

## Acknowledgements

CK is supported by the Helmholtz Infrastructure Initiative Frontiers in Arctic marine Monitoring (FRAM) and SH received support from the European Space Agency. SH was funded by the German Ministry of Economics and Technology (Grant 50EE1008). We thank Xuanji Wang and Yinghui Liu of the Cooperative Institute for Meteorological Satellite Studies (CIMSS), University of Wisconsin-Madison for their extensive work on the APP and APP-x climate data records, which was supported by the NOAA Climate Data Records Program. The APP-x Dataset is available online at ftp://data.ncdc.noaa.gov/cdr/appx/. Correspondence and requests for materials should be addressed to ckatlein@awi.de and to jeff.key@noaa.gov concerning the APP-x dataset. This study was funded by the Alfred-Wegener-Institut Helmholtz-Zentrum für Polar- und Meeresforschung. The views, opinions, and findings contained in this report are those of the authors and should not be construed as an official National Oceanic and Atmospheric Administration or U.S. Government position, policy, or decision.

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

**Figures**

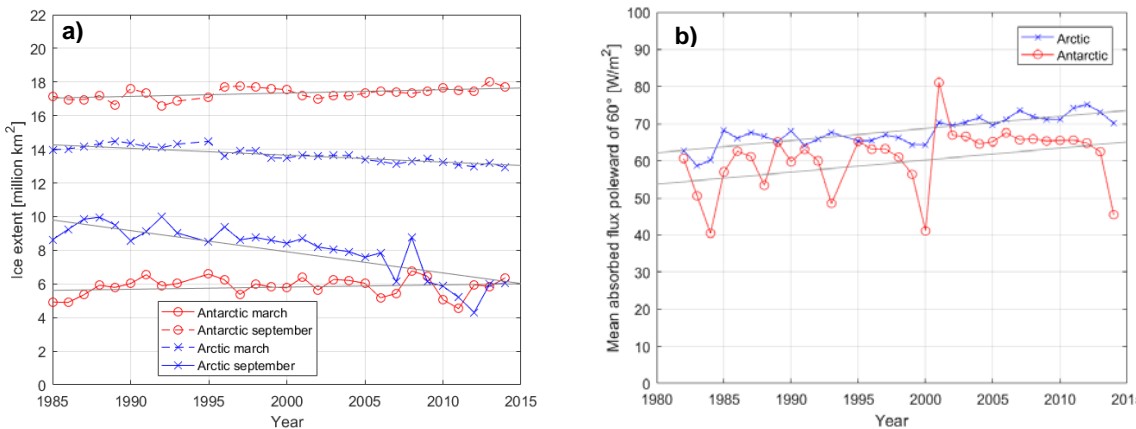

**Figure 1: a) Long-term trends of sea ice extent: Temporal evolution of sea ice extent in Antarctica (red) and the Arctic (blue) for minimal (solid line) and maximal (dashed line) seasonal extent as derived from APP-x data. Grey lines indicate fitted linear trends. b) Annual mean flux into ice ocean system: Shortwave radiative flux absorbed by the ice-ocean system poleward of 60° latitude in the Arctic (crosses) and Antarctic (circles).**

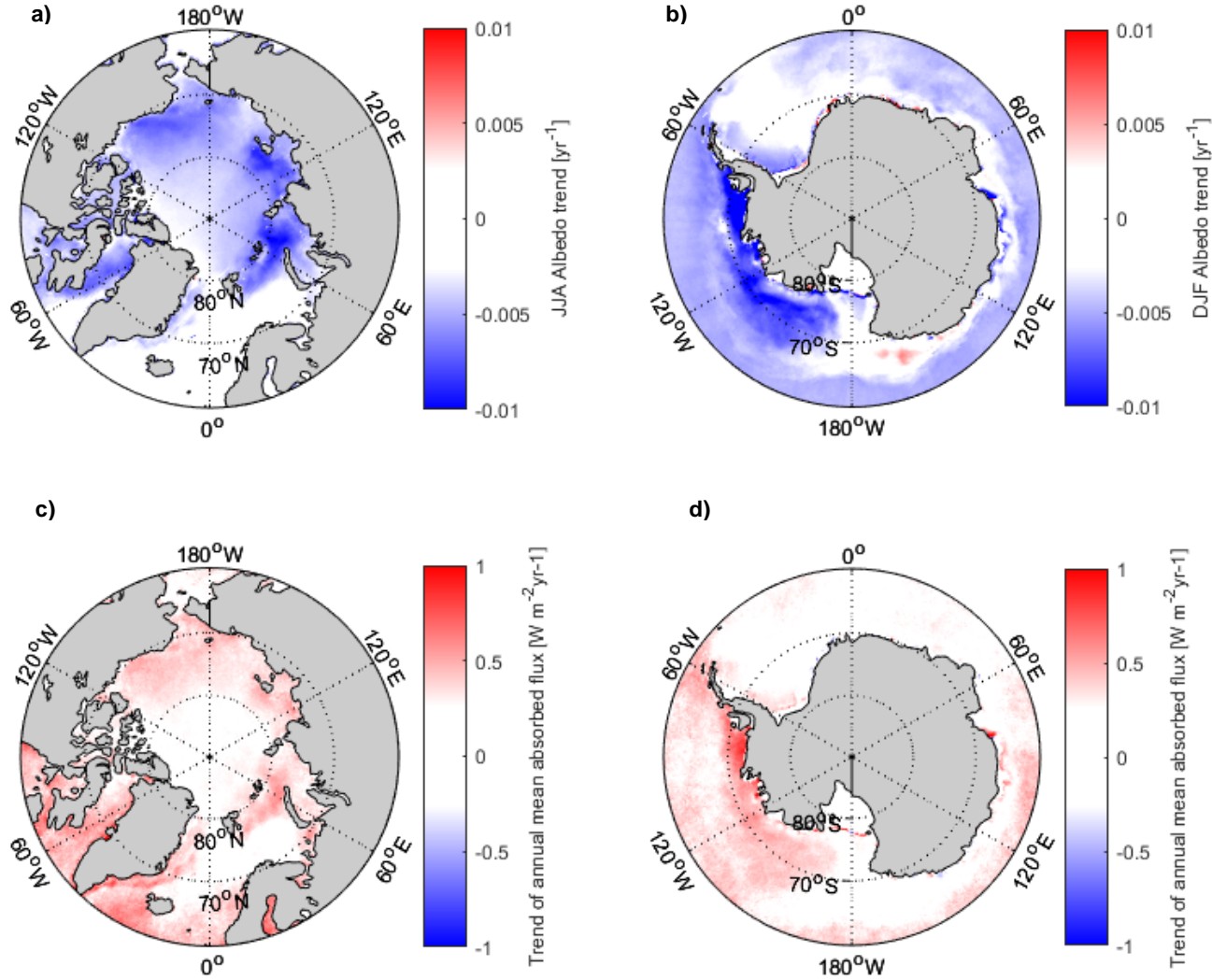

**Figure 2: Spatial distribution of trends in surface albedo and overall energy deposition into the polar oceans:** Trends [yr⁻¹] of mean daytime summer sea-ice albedo in the Arctic (a) and Antarctic (b) and trends [W/m²yr] of shortwave energy flux (c,d) absorbed by the ice-ocean system. The latter inherently includes both changes in albedo and cloud cover due to the nature of the APP-x dataset.

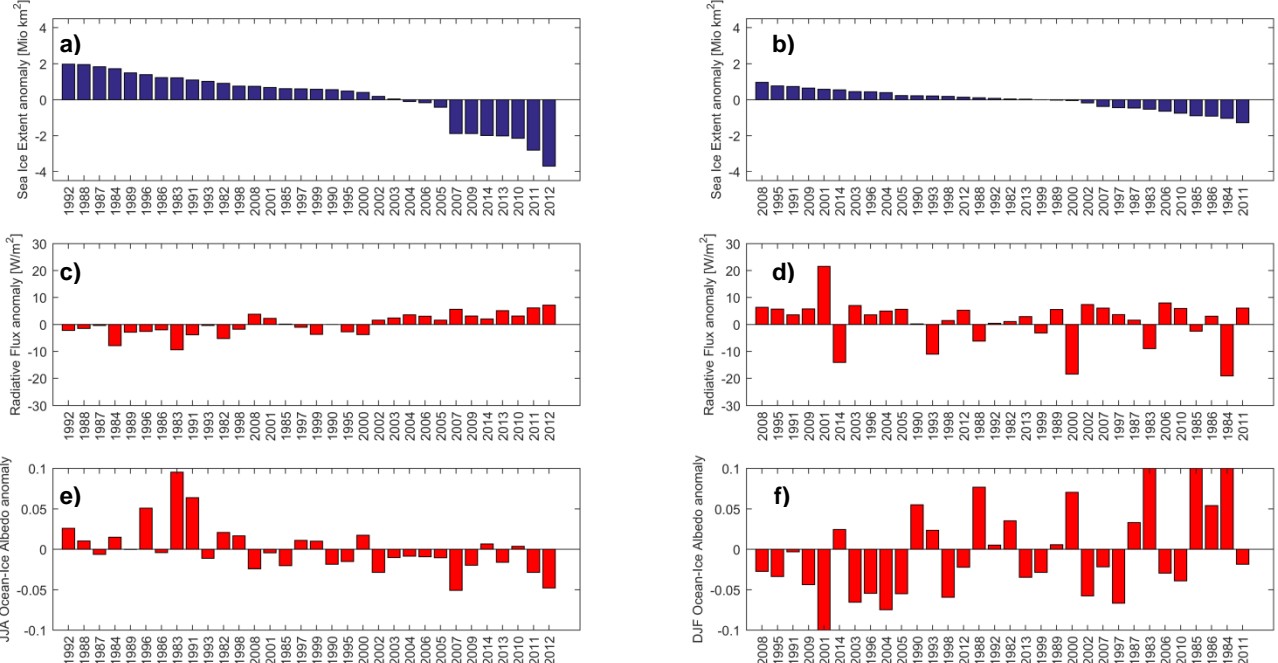

**Figure 3: Anomaly of mean, absorbed shortwave flux and surface albedo sorted from positive to negative sea-ice extent anomaly: Arctic (left) and Antarctic (right) sea ice extent anomaly (a,b), annual anomaly of shortwave flux absorbed by the ice ocean system polewards of 60° (c,d) and summer albedo anomaly (e,f).**

