# Peer review of "Brief communication: Increasing shortwave absorption over the Arctic Ocean is not balanced by trends in the Antarctic"

_The Cryosphere, 2016_

## Short Comment (SC1) · 8 Jan 2017

The values given in Figure 1 (b) are not correct. According to the CERES satellite data, from 2001-2014 the amount of sunlight absorbed into the arctic oceans is between 50 and 55 W/m2. Antarctic oceans varied between 55 and 58 W/m2 over the period.

The authors give values from 150 to 180 W/m2 for the Arctic and from 180 to 245 W/m2 for the Antarctic.

The Kiehl-Trenberth global energy budget gives a global average value of 167 W/m2 fro absorption, and the CERES data gives a global average of 162 W/m2 ... the authors' claim is larger than the K/T global average.

[Figure]

There is a separate issue. The CERES data shows a clear peak in Arctic oceanic absorption during the 2012 low in sea ice. There is no indication of this in the APP-x data.

Best regards,

w.

[Figure]

**Shortwave radiative flux absorbed by the ice-ocean
system poleward of 50° latitude
CERES EBAF Data**

Arctic
Antarctic
Global Average

**Fig. 1.**

[Figure]

**Average Absorbed Shortwave Flux**
**CERES EBAF Data, Mar-2000 to Feb-2015**
Avg Globe: 162.3  NH: 160.3  SH: 164.3  Trop: 214.3
Arc: 48.5  Ant: 35.2  Land: 137.8 Ocean: 171.8 W/m2

■ 20 W/m2   ■ 70 W/m2   ■ 120 W/m2   ■ 170 W/m2   ■ 220 W/m2   ■ 270 W/m2

**Fig. 2.**

**Average Absorbed Shortwave Flux Poleward of 50° Latitude**
CERES EBAF Data, Mar-2000 to Feb-2015
Avg Globe: 54.5  NH: 52.2  SH: 56.5  Trop: NaN
Arc: 45.3  Ant: 44.2  Land: NaN Ocean: 54.5 W/m2

| 20 W/m2 | 34 W/m2 | 48 W/m2 | 62 W/m2 | 76 W/m2 | 90 W/m2 |

**Fig. 3.**

---

## Referee Comment (RC1) · S. Bathiany (Referee) · 9 Jan 2017

General comments

In their brief communication, the authors focus on the surface short-wave balance in high latitudes. They use a new observational dataset (APP-x) to analyse trends over the recent decades and analyse the reasons behind these trends. I consider the format of a brief communication and the journal as a suitable choice to publish these results. Also, in the light of the large ongoing changes in the Arctic, the topic is of high relevance for science and the public. The general outline of the analysis is clear, and the relevant steps are explained.

[Figure]

However, I also see two major issues that in my opinion should be addressed before the paper can be published:

1. General motivation for this paper

I had some problems to understand the motivation behind this analysis. This is related to the fact that the authors aim to compare the hemispheres to each other. I am wondering why exactly the comparison of the hemispheres is useful or relevant. Is there anything we can learn about how the energy balance works? Or are there implications for predictions or other practical aspects? The analysis of trends in sea-ice albedo, sea-ice area (or extent) and cloud cover, and their combined effect on the short-wave (SW) absorption at the heart of this study. This is done individually for both hemispheres, and important differences are mentioned. The authors make the interesting point that SW absorption increases in the Southern Ocean, despite the slight increase in sea-ice area. To my taste, this could be communicated as the main result (if it is new) in title and abstract because it is much more straightforward than comparing hemispheres. Although the authors do not add energy balance terms from both hemispheres together, their wording sometimes suggests this. I suggest to not say that solar absorption in one hemishere "does not compensate" what is going on in the other hemisphere. I find this formulation confusing. I expect different effects at a certain location to be able to compensate each other, but not in two very remote regions that do not communicate directly. Another problem with the title is that it even suggests a comparison of two unrelated processes or units. This is of course not true but confronting "sea ice gain" in one hemisphere with "increased solar absorption" in the other hemisphere is unfortunate in my opinion. Alternatives for the title could be "Increasing short-wave absorption in southern high latitudes despite increasing sea ice area" or "Short-wave absorption is increasing over both of Earth's poles" or something similar. In general, the implications of this finding could be made clearer.

2. There seems to be a problem with the numbers, at least in Fig. 1b and the associated text. As pointed out in the interactive comment by W. Eschenbach, the short-wave

absorption values are not in agreement with other datasets. Some error might have been made in the calculations? The authors should correct it and check if their claims still hold.

Specific comments

1. Abstract: I think that it can be written more clearly (also see main comment 1).

2. Fig. 1: Also as pointed out by W. Eschenbach, the temporal evolution of the fluxes in Fig. 1b is surprising. I understand from Fig. 3 that despite the low sea ice in specific years, we cannot automatically assume a large peak in SW absorption in these years, mainly because of confounding effects of cloud cover fluctuations? However, I am also wondering about the peculiar oscillation-like pattern in the Arctic time series, with a rapid increase every 5 years, followed by an accelerating decrease. This pattern is repeated several times with increasing magnitude. The authors may want to check if this behaviour is real.

3. Fig. 3: The peaks in Fig. 3c seem to not coincide with those in Fig. 1b, or are at least shifted in time. For example, in Fig. 1b, 2000 is a year with low absorption in the Arctic and 2001 is a year with high absorption. But in Fig. 3c, both years have negative anomalies, whereas 2002 has a positive anomaly. Also, year 1994 is missing. I understand from the text why it is missing in the southern hemisphere, but it could be included in the analysis of the northern hemisphere.

4. What is the reason for the increased downwelling SW radiation over the Southern Ocean? The authors could elaborate a bit on the role of cloud cover. Is this signal expected in the light of anthropogenic climate change, or is it random (internal variability)? So, what could be expected for the future?

5. Methods: Why is it necessary to first calculate the cumulative absorbed energy in the whole region over one year and then convert it into fluxes again for the figures? The dataset already seems to contain fluxes. The potential calculation error mentioned
above may be related to this. Is there any good reason why the sea ice extent is defined differently than in other studies?

---

## Referee Comment (RC2) · Anonymous Referee #2 · 8 Feb 2017

The Brief Communication uses a new dataset, APP-x, to examine trends in the annual mean absorbed flux of shortwave radiation in the polar regions from 1985-2014. It addresses relevant scientific questions within the scope of the journal. The conclusions are appropriate for a Brief Communication. The article is clearly written and easy to read. However I have some criticisms.

I felt the authors could improve the description of the motivation for the study. For example, why consider only shortwave radiation and not include the emitted longwave radiation? The stated goal is "to determine to what extent the increased absorption of solar shortwave energy caused by losses in Arctic summer sea ice can compensate for the decreased absorption caused by modest increases of sea ice extent in the

Antarctic." But why? I assume the reason for doing this is to assess the contribution to the overall global energy balance? If so, please comment on how you think how longwave radiation would contribute to the story?

Commas are placed in unusual places and sometimes seem to be missing; eg p. 1, line 9 insert comma after "sea-ice". Please check throughout the document.

p. 1, Line 19: Please give a reference for the debate.

p. 2: I think that the AAP-x data set calculates ice thickness using a 1D OTIS model with satellite-derived input variables. Is there any validation of these AVHRR ice thicknesses in the Southern Hemisphere?

p. 2, Line 22: More snow on Antarctic sea ice is also likely to give a higher albedo in the Southern Ocean.

p. 3: It is excellent that the authors compare sea ice concentration derived from the APP-x data with passive microwave concentration data. But I would like to have seen some sort of quantitative comparison. This would increase the reader's confidence in the new dataset.

p. 3, Lines 10-15: I would like to have seen some estimate of the error in the quoted measurements.

p. 3, Line 26-27: I did not understand this sentence. Do you mean "this does not significantly affect our analysis of energy fluxes, as the largest uncertainty in the albedo occurs with low fluxes, subsequently leading to a low uncertainty in the time-averaged energy flux."?

p. 3, Line 29: ahha – here snow cover is mentioned. Please support with a reference.

p. 4, line 1-2: Is the APP-x surface albedo not used to obtain ice thickness through the OTIM model? How does the lack of independence of surface albedo and ice thickness affect the calculation of shortwave energy flux?

p. 4, Line 3-4: Do you really know the annual shortwave energy flux to 0.1 in 200 Wm-2? Please make it clear that you have considered the accuracy of your results.

p. 4, Line 6-7 states that "Average Southern Hemisphere absorption remained relatively constant throughout the satellite record" while on line 13 it states "In the Southern Ocean energy absorbed by the ice-ocean system south of 50oS also increased . . ." I am confused by this apparent contradiction and I suspect that I have missed a subtlety.

p. 4, Line 12-13: I think you say the total annual shortwave energy in the northern hemisphere increases at a significant rate, while in the Southern hemisphere it did not. Yet the numbers seem similar (8.77 X 1025 Jyr-1 compared with 6.14 X 1025 Jyr-1). Please justify. Again I may have missed a subtlety but, if so, perhaps you could make this clearer.

Fig 1a): Confidence would be increased in the dataset if the slope from another reliable source (eg NSIDC) was added to the figure.

Fig 1b): What are the intriguing jumps in the data?

Fig 2: The trend should be in Wm-2yr-1. Remove comma after "both"

Fig 3: Caption and figure do not agree.

---

## Author Comment (AC1) · 25 Apr 2017

**Replies to Reviewer Comments: Reviewer 1 (W. Eschenbach):**

Please find our replies to the reviewer's comments in blue print.

The values given in Figure 1 (b) are not correct. According to the CERES satellite data, from 2001-2014 the amount of sunlight absorbed into the arctic oceans is between 50 and 55 W/m2. Antarctic oceans varied between 55 and 58 W/m2 over the period.

The authors give values from 150 to 180 W/m2 for the Arctic and from 180 to 245 W/m2 for the Antarctic.

The Kiehl-Trenberth global energy budget gives a global average value of 167 W/m2 fro absorption, and the CERES data gives a global average of 162 W/m2 ... the authors' claim is larger than the K/T global average.

We thank you very much for bringing up this point. Indeed there was a mistake in our calculations. Cells with invalid retrievals are labelled as NaN (not-a-number) in the APP-x product. NaN cells were ignored during averaging, resulting in numbers that were too large in our manuscript. We fixed this by attributing zero fluxes to NaN values, as incorrect retrievals originate from too little sunlight in winter. Consequently we set NaN albedos to 1. Our numbers lie now in the range that you report, considering the recently identified underestimation of absorbed fluxes in the CERES data over the Arctic (Riihelä 2017, JGR, in press).

However you unfortunately did not provide any sources or calculation details for your numbers. Your plots are rather inconsistent regarding latitude limits. In these calculations the threshold latitude is of significant importance. While you label all your plots as "absorption poleward of 50°", your second plot clearly only shows data starting from 60°. Also the dotted lines in your plot (~66°) probably present what your plotting routines define as "Arctic" and "Antarctic". We can just assume that your first plot also just shows data poleward of 66°, as even your black "global average" line is around 54 W/m² despite the fact that you claim above a global average of ~165 W/m²

Of course lower latitude contributions to the mean value are crucial due to their relatively large areas and larger fluxes. For a more poleward latitudinal threshold smaller numbers result as average values.

This boils down to a central question for this study: What is the right cut-off latitude? The latitudinal extent of sea-ice differs dramatically both between the hemispheres and also within the different regions of the hemispheres themselves. Our initial approach was to encompass all sea-ice for our calculations. However, we see that this might be confusing as it differs from generally used classifications. Also this leads to a high number of processes other than sea-ice retreat influencing the results.

We thus recalculated for a latitudinal threshold of poleward of 60°. While this still encompasses some of the generally ice-free oceans, it covers most of the sea-ice extent throughout all seasons. We do not want to reduce this limit further to 66° as this would exclude a significant portion of the area seasonally covered by sea ice and areas that have experienced large changes in sea ice cover in recent decades.

The APP-x shortwave fluxes have been successfully validated e.g. against observations from SHEBA, a drifting observation ice camp, as well as again CERES fluxes (Riihelä 2017, JGR in press). We are thus confident in the calculated fluxes.

There is a separate issue. The CERES data shows a clear peak in Arctic oceanic absorption during the 2012 low in sea ice. There is no indication of this in the APP-x data.

This is also visible in the APP-X data after correcting the calculations. A new version of Figure 1b will be included in the revised version.

---

## Author Comment (AC2) · 25 Apr 2017

**Replies to Reviewer Comments: Reviewer 2 (S. Bathiany):**

Please find our replies to the reviewer's comments in blue print.

General comments

In their brief communication, the authors focus on the surface short-wave balance in high latitudes. They use a new observational dataset (APP-x) to analyse trends over the recent decades and analyse the reasons behind these trends. I consider the format of a brief communication and the journal as a suitable choice to publish these results. Also, in the light of the large ongoing changes in the Arctic, the topic is of high relevance for science and the public. The general outline of the analysis is clear, and the relevant steps are explained.

We thank you very much for the positive evaluation of our work. We are happy that you also consider a brief communication in "The Cryosphere" as the right outlet for this research.

However, I also see two major issues that in my opinion should be addressed before the paper can be published:

1. General motivation for this paper

I had some problems to understand the motivation behind this analysis. This is related to the fact that the authors aim to compare the hemispheres to each other. I am wondering why exactly the comparison of the hemispheres is useful or relevant. Is there anything we can learn about how the energy balance works? Or are there implications for predictions or other practical aspects? The analysis of trends in sea-ice albedo, seaice area (or extent) and cloud cover, and their combined effect on the short-wave (SW) absorption at the heart of this study. This is done individually for both hemispheres, and important differences are mentioned. The authors make the interesting point that SW absorption increases in the Southern Ocean, despite the slight increase in sea-ice area. To my taste, this could be communicated as the main result (if it is new) in title and abstract because it is much more straightforward than comparing hemispheres. Although the authors do not add energy balance terms from both hemispheres together, their wording sometimes suggests this. I suggest to not say that solar absorption in one hemishere "does not compensate" what is going on in the other hemisphere. I find this formulation confusing. I expect different effects at a certain location to be able to compensate each other, but not in two very remote regions that do not communicate directly.

Indeed our motivation for this study is mainly coming from requests from the public (and media) that we frequently get. We agree that the concept of putting both hemispheres in one pot is somewhat far-fetched in a scientific way of thinking alone. However, this comparison is frequently made by "climate sceptics", where increases in Antarctic sea ice cover are used as an argument to counter the decreasing trend in Arctic sea ice. We feel the urgent need to address this through an analysis of not just the changes in sea ice in the two polar regions, but also the impact of those changes on the surface shortwave radiation budget, which in turn impacts sea ice. Furthermore, the discussion paper has received significant attention in the community (article metrics) and related media requests show that the comparison of both hemispheres is a relevant topic. Having said that, the reviewer makes a good point that the finding of increased absorbed solar radiation in the Antarctic is interesting and important. We will emphasize this point in the revised manuscript.

Another problem with the title is that it even suggests a comparison of two unrelated processes or units. This is of course not true but confronting "sea ice gain" in one hemisphere with "increased solar absorption" in the other hemisphere is unfortunate in my opinion. Alternatives for the title could be "Increasing short-wave absorption in southern high latitudes despite increasing sea ice area" or "Short-wave absorption is increasing over both of Earth's poles" or something similar. In general, the implications of this finding could be made clearer.

We agree with your view that the title is comparing apples and oranges somewhat, so we suggest an alternative title combining your suggestions "Increasing short-wave absorption over the sea ice area at both poles"

2. There seems to be a problem with the numbers, at least in Fig. 1b and the associated text. As pointed out in the interactive comment by W. Eschenbach, the short-wave absorption values are not in agreement with other datasets. Some error might have been made in the calculations? The authors should correct it and check if their claims still hold.

Indeed there was a mistake in our calculations. Cells with incorrect retrievals are labelled with NaN in the App-X product. Unfortunately NaN cells were ignored during averaging, leading to way to high numbers in our manuscript. We fixed this by attributing zero fluxes to NaN values, as incorrect retrievals originate from too little sunlight in winter. Consequently we set NaN albedos to 1. Our numbers lie now in the range cited by W. Eschenbach, considering the known underestimation of absorbed fluxes by CERES data (Riihelä 2017, JGR, in press). However our conclusions stay unaffected by this calculation error.

Specific comments

1. Abstract: I think that it can be written more clearly (also see main comment 1).

We reformulated the abstract in the revised version

2. Fig. 1: Also as pointed out by W. Eschenbach, the temporal evolution of the fluxes in Fig. 1b is surprising. I understand from Fig. 3 that despite the low sea ice in specific years, we cannot automatically assume a large peak in SW absorption in these years, mainly because of confounding effects of cloud cover fluctuations? However, I am also wondering about the peculiar oscillation-like pattern in the Arctic time series, with a rapid increase every 5 years, followed by an accelerating decrease. This pattern is repeated several times with increasing magnitude. The authors may want to check if this behaviour is real.

We thank you for pointing out the oscillation-like pattern with jumps. As this stayed visible also after accounting for NaN retrievals and changing the cut-off latitude, we found that these patterns are likely related to drifting equator crossing times during the lifetime of individual satellites. The App-X radiative flux product is already corrected for this bias but somehow some bias remains. We will investigate this potential bias further and try to correct it in the revised version.

3. Fig. 3: The peaks in Fig. 3c seem to not coincide with those in Fig. 1b, or are at least shifted in time. For example, in Fig. 1b, 2000 is a year with low absorption in the Arctic and 2001 is a year with high absorption. But in Fig. 3c, both years have negative anomalies, whereas 2002 has a positive anomaly. Also, year 1994 is missing. I understand from the text why it is missing in the southern hemisphere, but it could be included in the analysis of the northern hemisphere.

This seems to be due to the drifting equator-passing times (see above comment). We updated the description in the text. Actually the data outage also affects the northern Hemisphere in 1994.

4. What is the reason for the increased downwelling SW radiation over the Southern Ocean? The authors could elaborate a bit on the role of cloud cover. Is this signal expected in the light of anthropogenic climate change, or is it random (internal variability)? So, what could be expected for the future?

We added some discussion on the reasons for the increased downwelling SW radiation. This seems to be caused by cloud cover. However one would rather expect an increasing cloud cover with retreating ice, thus we speculate that this is driven by local effects and interannual variability.

5. Methods: Why is it necessary to first calculate the cumulative absorbed energy in the whole region over one year and then convert it into fluxes again for the figures? The dataset already seems to contain fluxes. The potential calculation error mentioned above may be related to this.

Of course flux averages are calculated directly on the fluxes given by the dataset. We clarify this in the revised version.

Is there any good reason why the sea ice extent is defined differently than in other studies?

We added some explanations as to why we used this threshold of ice-thickness bigger than zero. APP-x does not contain any ice-concentration field and to stay consistent we did not want to use any alternative source. However the APP-x dataset uses NSIDC ice concentration >15% internally during the retrieval to determine whether a pixel is ice-covered. Thus all pixels with sea ice concentration > 15% have a valid ice-thickness and pixels below the threshold don't. So we are indeed using the same definition as in other studies. Differences to other studies just result from the rather course resolution of APP-x

---

## Author Comment (AC3) · 25 Apr 2017

**Replies to Reviewer Comments: Anonymous Referee #2**

The Brief Communication uses a new dataset, APP-x, to examine trends in the annual mean absorbed flux of shortwave radiation in the polar regions from 1985-2014. It addresses relevant scientific questions within the scope of the journal. The conclusions are appropriate for a Brief Communication. The article is clearly written and easy to read. However I have some criticisms.

We thank you very much for the constructive evaluation of our work!

I felt the authors could improve the description of the motivation for the study. For example, why consider only shortwave radiation and not include the emitted longwave radiation? The stated goal is "to determine to what extent the increased absorption of solar shortwave energy caused by losses in Arctic summer sea ice can compensate for the decreased absorption caused by modest increases of sea ice extent in the Antarctic." But why? I assume the reason for doing this is to assess the contribution to the overall global energy balance? If so, please comment on how you think how longwave radiation would contribute to the story?

Indeed our motivation for this study is mainly coming from requests from the public (and media) that we frequently get. We agree that the concept of putting both hemispheres in one pot is somewhat far-fetched in a scientific way of thinking alone. However this comparison is frequently made by "climate sceptics", so we feel the urgent need to counter this with a reasonable scientific answer. Furthermore, the discussion paper has received significant attention in the community (article metrics) and related media requests show that the comparison of both hemispheres is a relevant topic.

We added some discussion on longwave radiation. However we want to keep the manuscript focused on shortwave and do not attempt to describe the full energy balance of the polar regions as this would be too large for a brief communication. We focus on only shortwave because besides its major role in sea ice melt (solar heating & melting) it also impacts many other aspects of the sea ice system, such as biological production.

Commas are placed in unusual places and sometimes seem to be missing; eg p. 1, line 9 insert comma after "sea-ice". Please check throughout the document.

We will check the manuscript for typographic errors throughout the text.

p. 1, Line 19: Please give a reference for the debate.

We reformulated for clarity, as this is not really a debate.

p. 2: I think that the AAP-x data set calculates ice thickness using a 1D OTIS model with satellite-derived input variables. Is there any validation of these AVHRR ice thicknesses in the Southern Hemisphere?

We agree, that ice thickness is hardly validated due to a lack of reference data. However ice thickness was only used for ice detection, so it does not impact our calculations.

As an aside, the one-dimensional thermodynamic ice model (OTIM) is described in Wang, X., J. Key, and Y. Liu, 2010, A thermodynamic model for estimating sea and lake ice thickness with optical satellite data, J. Geophys. Res.-Oceans, 115, C12035, doi:10.1029/2009JC005857.)

p. 2, Line 22: More snow on Antarctic sea ice is also likely to give a higher albedo in the Southern Ocean.

Yes, but snow on sea-ice is not separately accounted for in the APP-x dataset. We added another mention of the effect of snow on albedo in the revised version

p. 3: It is excellent that the authors compare sea ice concentration derived from the APP-x data with passive microwave concentration data. But I would like to have seen some sort of quantitative comparison. This would increase the reader's confidence in the new dataset.

As the ice thickness retrieval is thresholded internally by the NSIDC ice concentration, our APP-x ice extent is effectively a regridded NSIDC ice extent. Thus we do not see the need for a further comparison. For your reference, we provide a version of figure 1A with passive microwave ice extent from the University of Bremen at the end of this document.

p. 3, Lines 10-15: I would like to have seen some estimate of the error in the quoted measurements.

We added confidence intervals as determined during the trend-fitting to all numbers.

p. 3, Line 26-27: I did not understand this sentence. Do you mean "this does not significantly affect our analysis of energy fluxes, as the largest uncertainty in the albedo occurs with low fluxes, subsequently leading to a low uncertainty in the time-averaged energy flux."?

Yes, we rephrased accordingly to make it clearer

p. 3, Line 29: ahha – here snow cover is mentioned. Please support with a reference.

We added a reference.

p. 4, line 1-2: Is the APP-x surface albedo not used to obtain ice thickness through the OTIM model? How does the lack of independence of surface albedo and ice thickness affect the calculation of shortwave energy flux?

We agree, but as we did not split solar energy deposition into ice and ocean compartments all that counts is albedo. Albedo retrieval is independent of the ice-thickness retrieval, while the modelled ice-thickness is of course dependent on the retrieved albedo.

p. 4, Line 3-4: Do you really know the annual shortwave energy flux to 0.1 in 200 Wm-2? Please make it clear that you have considered the accuracy of your results.

Thank you for pointing out this inconsistency. While these results are numerically correct according to the APP-x product, the accuracy of the entire measurement is of course less. We thus rounded these numbers to the significant digits in the revised version.

p. 4, Line 6-7 states that "Average Southern Hemisphere absorption remained relatively constant throughout the satellite record" while on line 13 it states "In the Southern Ocean energy absorbed by the ice-ocean system south of 50oS also increased : : :" I am confused by this apparent contradiction and I suspect that I have missed a subtlety.

We reformulated for clarity and corrected the wrong exponent (see below).

p. 4, Line 12-13: I think you say the total annual shortwave energy in the northern hemisphere increases at a significant rate, while in the Southern hemisphere it did not. Yet the numbers seem similar (8.77 X 1025 Jyr-1 compared with 6.14 X 1025 Jyr-1). Please justify. Again I may have missed a subtlety but, if so, perhaps you could make this clearer.

Thank you for pointing this error out. It was of typographic nature as the number in the exponent was wrong. We corrected this in the revised version.

Fig 1a): Confidence would be increased in the dataset if the slope from another reliable source (eg NSIDC) was added to the figure.

As explained above, the data is derived using a widely accepted NSIDC product, so we do not see the need for further comparison. Adding further lines into the figure would confuse it. For your reference we just provide a version of the figure with black dotted / dash-dotted lines representing the according ice extent data from the University of Bremen below. As mentioned in the text, the magnitude is affected due to the different resolution, but the trends are very similar.

[Figure]

Fig 1b): What are the intriguing jumps in the data?

We thank you for pointing out these jumps. As this stayed visible also after accounting for NaN retrievals and changing the cut-off latitude, we found that these patterns are related to drifting equator crossing times during the lifetime of individual satellites. Thus we introduced an additional correction scaling the retrieved fluxes according to solar zenith angle during the actual satellite overpass. We rescaled the fluxes to zenith angles corresponding to 0:00 and 12:00 local solar time to achieve best possible estimates of daily mean fluxes.

Fig 2: The trend should be in Wm-2yr-1. Remove comma after "both"

Corrected accordingly

Fig 3: Caption and figure do not agree.

Corrected accordingly

---

## Referee Report (RR1)

Review on "Brief communication: Increasing shortwave absorption over the sea ice area at both poles"

General comments

In my opinion, the manuscript has improved in readability compared to the previous submission. The authors have made some statements clearer and have corrected shortcomings in their analysis.

I still have some concerns regarding the relatively small amount of robust and novel conclusions in this paper. In the paper, and even more so in the discussion, the authors mention the interest of the media as well as claims by so-called climate sceptics as a motivation for their study. However, a research paper is primarily meant to address other researchers. The main statement that the short-wave absorption is increasing should hence be complemented with an analysis of the causes. In my opinion the article could therefore elaborate a bit more on the potential scientific merits of its approach, and explore the main statements in some more analytical detail.

Specifically, the conclusions about the role of sea-ice albedo and clouds could be supported with some more evidence. From the analysis of sea ice distribution, surface albedo and absorbed short-wave radiation, the authors conclude that changes in cloud properties and ice albedo are responsible for the observed changes in short-wave absorption, but these properties are not shown directly. It would be ideal if the authors could somehow quantify the individual contributions of changes in sea-ice albedo and cloud coverage in order to support their statements. For example, how has the downwelling short-wave flux changed? As several cloud properties are also included in APP-x (for example cloud radiative forcings), it might not be out of proportion for this brief communication to test whether changes in these properties are in line with the author's claims that are based on some other variables of the same dataset.

In the conclusions, the authors repeat their statement that the short-wave absorption is increasing. I would also expect some statements about the scientific relevance of this finding, e.g. implications about our understanding of the polar climate, and how future research should address the questions that have to remain open.

Specific comments

- p. 2, l. 14: It is stated here and also at other places (e.g. p. 4, l. 13/14) that the dataset "takes into account" changes in cloud cover, albedo changes, etc. This seems to relate to the statement that "the energy balance is kept closed" (p. 2, l. 22/23) which seems to involve some kind of model. I suggest to explain this in more detail. Why is it an advantage to use this approach instead of independent observations of different variables? For readers not familiar with the available data, it could be explained in more detail which properties are observed and which are inferred by the authors involving additional assumptions.
- p. 3, l. 8: "accounts for true cloud cover". How? And what is "true"?
- Fig. 1: Why is the short-wave absorption much more variable in the Southern hemisphere? Given these fluctuations, is the trend significant at all? If not, it should not be claimed that the solar absorption has been increasing, and the conclusions might be better based on the correlations between variables instead of the trends.
- Fig. 2: I am puzzled by the fact that the sign of the trends is spatially so uniform, even in places where there is no sea ice. How can this be the case? Could there be a bias?

Minor comments

Main text

- What is new about the APP-x dataset, i.e. what makes it a "novel tool"? Hasn't AVHRR data been available for a long time?
- p. 2, l. 12: remove "large"
- p. 3, l. 11: "slightly larger numbers" – how much larger approximately?
- p. 4, l. 3: rephrase to make clearer: "seasonal Arctic sea ice..., the dataset shows a decrease..., while in the Antarctic, albedo trends show regional differences ... ice concentration."
- p. 4, l. 11: remove "relatively"
- p. 4, l. 15: "The mean annual shortwave energy flux...."
- p. 4, l. 18/19: Why is the absorption more spatially uniform in the Arctic?
- p. 4, l. 24: I suggest to cite the source of this result, or explain how this test was done.
- p. 4, l. 26/27: "energy absorption" – isn't it only the short-wave absorption and not the full energy balance what the authors address here?
- p. 4, l. 30: "does not show a pattern" – I don't understand what is meant here.
- p. 5, l. 11: "Increases in Antarctic sea ice only occur during the Southern Hemisphere winter". Fig. 1a seems to contradict this statement. Anyways, I do not consider this aspect important enough for the conclusions section. Instead, I would prefer to see a discussion of the implications of the main results (see above).

Fig. 1
- Fig. 1: The caption refers to minimal and maximal seasonal extent, but within Fig. 1a it is referred to March and September (by the way, months are written with a capital letter).

Fig. 2

- Caption: trends in "sea ice albedo". Is this really what the figure is showing? As far as I understand it shows the surface albedo change.
- I suggest to use a different colour bar that gives less weight to very small changes and more weight to large changes.
- a) and b) "Arctic" and "Antarctic" are unnecessary details in the variable name because one can see what region it is on the map anyways.

Fig. 3
- This figure is not discussed in the text. I suggest to embed it in the line of argument more explicitly.
- "albedo anomaly". Surface albedo, I guess?!

---

## Author Response (AR2)

**Replies to Review comments:**

We thank you very much for your efforts in evaluating our work. Please find our replies in blue font under your comments.

**Editor's comments:**

I have now received two reviews of the revised version of your manuscript, see attached. You will find that both reviewers agree on a crucial point, namely that your numbers suggest an insignificant increase in Antarctic SW absorption. They hence find that the current title and abstract of the manuscript are misleading.

We agree that the title may be slightly misleading. However, this title was a result of the previous review round and a direct suggestion of the reviewers. We returned to a title close to our initial submission: "Increasing shortwave absorption over the Arctic Ocean is not balanced by trends in the Antarctic". We still consider this a very important result that has not been previously shown in the literature. We have reformulated the text to stress the main finding of similar trends in direction and magnitude in both hemispheres. The Antarctic trend is just masked in a larger interannual variability leading to statistical insignificance at 95% level (it would be significant at 90% level).

The reviewers also agree that the general result for the Arctic is certainly not surprising, which poses some doubts regarding the amount of truly novel information contained in this manuscript.

As indicated in our manuscript, this is the first study to make a quantitative comparison on a consistent bi-hemispheric observational dataset. Furthermore, these concerns are only raised by referee #1 (Sebastian Bathiany), while referee #3 (Ian Eisenman, report #2) does not mention these concerns, but rather states "The manuscript is clearly written, and I find it well suited for publication in this journal" and evaluates "accepted subject to minor revisions" not encompassing further analysis.

I tend to agree with the view of reviewer 1 that the overall scientific content of this manuscript must be increased for it to become publishable in *The Cryosphere*. This is also reflected by the concerns of reviewer 2 who finds the result for the Arctic somewhat trivial and the significance of the result for the Antarctic unclear. Reviewer 1 poses some suggestions to improve the significance of your study, which I hope you will find helpful for revising this manuscript.

As mentioned above, referee #3 (Ian Eisenman, report #2) only mentions that the results for the Arctic are not surprising. However, there is no indication by him that the Arctic results are the main part of the manuscript. He explicitly states "The manuscript is clearly written, and I find it well suited for publication in this journal."

All reviewers agreed on the format of a brief communication in *The Cryosphere* (Ian Eisenman in this review round: "… well suited for publication in this journal" and Sebastian Bathiany during the first round of reviews: "I consider the format of a brief communication and the journal as a suitable choice to publish these results."). We are firm on our decision not to further extend the analysis but to use this study to stimulate further scientific work. The tasks necessary to add additional analysis go well beyond the scope of a "Brief communication" and resources for such an analysis are lacking.

For such revised version, I in particular ask that you more explicitly specify the scientific advance of your manuscript for the scientific audience of The Cryosphere.

We tried to reformulate some passages to make this clearer. We wrote this paper because there is so far no scientific literature covering the combined effects of Arctic and Antarctic shortwave energy balance, which we consider an important topic. Changes in the two polar regions have not always been in the same direction, e.g., dramatic decreases in Arctic sea ice extent countered by increases in the Antarctic sea ice cover. But what do these often opposite changes mean in terms of the surface energy budget? We consider it important to provide scientific evidence to questions that are raised by society and to stress the importance of a global view on sea ice process related studies.

**Comments by Referee #3 (Ian Eisenman)**

This Brief Communication assesses the change in surface shortwave absorption in both Polar Regions during 1982-2014. The authors find an increase in the Arctic, which is unsurprising given the rapid sea ice retreat, but they also report an increase in the Antarctic despite the observed sea ice expansion in that hemisphere. The manuscript is clearly written, and I find it well suited for publication in this journal, although I think the points raised below should first be addressed.

We thank you very much for your positive evaluation of our work! Indeed, the increase in the Arctic is not surprising, as apparent also from the many citations given in the text. What is important here is how the increase in the Arctic compares to the trend in absorption in the Antarctic, particularly given the different trends in sea ice. We are happy that you positively evaluate our main finding that the slight increase in Antarctic shortwave absorption (or a statistically significant non-decrease) does not compensate for Arctic increases.

Major comment:

I am a bit confused regarding the significance of the Southern Hemisphere result. The authors write that in the Southern Hemisphere "energy absorption by the ice-ocean system south of 60S also increased but, due to the large interannual variability, only at a statistically insignificant rate" (page 4 lines 23-25). But despite this, the text says "both hemispheres show a distinct increase in energy deposition" (page 4 line 25) and "positive energy absorption trends in both hemispheres" (page 5 line 1), the title states that shortwave absorption is increasing "at both poles", and the abstract says it is "increasing at both poles". Am I missing something here? If the results do not show a significant increase in shortwave absorption in the Southern Hemisphere, then I think the title, abstract, and main text should be revised to reflect this point.

We thank you very much for highlighting this inconsistency. This title was suggested by the other reviewer during the first round of revisions. We agree with your concerns and returned to a title (as well as some text formulations) closer to our initial submission. Also, the lack of clarity may have come from the fact that the shortwave absorption trends are actually very similar in magnitude, but the large interannual variability in the Antarctic renders the southern trend statistically insignificant. We however still find this an important result and we reformulated this for clarity.

Minor comments:

page 4 line 22-23: I found the choice of units of the increase in absorbed shortwave to be somewhat unclear. It seems most intuitive to present this in terms of how much the absorbed energy in W/m^2 (averaged over the relevant polar region) increases each year. This is instead presented here in units of J/yr, which I find somewhat confusing. Furthermore, I am confused by the values. The rate of increase reported here for the region 60N-90N is 2.3 x 10^25 J/yr (page 4 line 22-23). But unless I'm mistaken, this is more than the total TOA solar energy incident on the earth, which is 5.5 x 10^24 J/yr (calculated as 1340 W/m^2 * surface area of earth/4).

As the areal distribution of the sea ice zone is significantly different at both poles, and we wanted to provide a measure that allows us to easily compare the budget of both hemispheres, we chose J/yr. We agree that the presented numbers were too high. We found an error in our calculations and now the numbers are 5 orders of magnitude lower. The correction of this error however does not have an effect on our main conclusion. Thank you very much for catching this mistake!

page 3 line 25: The 1982-2004 sea ice trends reported in this paragraph, averaged between September and March, have a magnitude in the Arctic that is 5.2x larger than in the Antarctic. This is fairly similar to the 1979-2012 annual values reported in the IPCC AR5, which have a magnitude in the Arctic that is 2.9x larger than in the Antarctic. I think it would be more accurate to revise how this is described from Arctic losses being "thus roughly one order of magnitude stronger" than Antarctic increases to "thus five times larger in magnitude" (or something similar).

We thank you very much for your suggestion and adopted your suggested formulation.

page 3 line 21: When error bars are given for trends, are these 68% confidence intervals (as are often used), or are they 95% confidence intervals (as implied on page 3 line 19)? It would be helpful if this were specified.

Indeed, these are 95% confidence intervals. We added the following statement to the preceding paragraph: "Given error intervals for trends are 95% confidence intervals."

page 1 line 19: The argument being made here may be supported by our own finding that there is a large uncertainty in the 1979-2012 Antarctic sea ice extent trend due to intercalibration across sensor changes (Eisenman et al. 2014, The Cryosphere 8 1289-1296). I don't mean to angle for a self-citation here, and I certainly leave this to the authors' discretions, but I just figured I'd mention this in case the authors find that it strengthens their case.

We agree with you on this point! This work was referenced in an earlier version of the manuscript but removed due to the comments during editorial review. We have added it again, as it is an important example showing inconsistencies in the datasets.

page 2 lines 21-29: (typo) ice extent trends should be in units of km^2/yr, not km/yr.

Maybe there was some error with the PDF display? We double checked that the appropriate units are used.

page 2 lines 26-27: (typo) "as a climate data record variables" should presumably be "as climate data record variables".

Corrected

page 3 line 22: (typo) "in the Antarctic in the Southern Hemisphere" should presumably be "in the Southern Hemisphere".

Corrected

**Comments by referee #1 (Sebastian Bathiany)**

 General comments
In my opinion, the manuscript has improved in readability compared to the previous submission. The authors have made some statements clearer and have corrected shortcomings in their analysis.

We thank you for the positive evaluation of our revised version.

I still have some concerns regarding the relatively small amount of robust and novel conclusions in this paper. In the paper, and even more so in the discussion, the authors mention the interest of the media as well as claims by so-called climate sceptics as a motivation for their study. However, a research paper is primarily meant to address other researchers.

It is a valid objective of a scientific publication to raise questions and to stimulate scientific discussion based on findings in a novel observational data record, especially since we chose the format of a Brief Communication and not a full-length paper. We would not have submitted this paper if the information presented was covered in existing scientific literature. Also, science may very well be driven by the curiosity of society, so we think that if questions arise from society in public discussion, they actually should be appropriately addressed in the scientific literature.

The main statement that the short-wave absorption is increasing should hence be complemented with an analysis of the causes. In my opinion the article could therefore elaborate a bit more on the potential scientific merits of its approach, and explore the main statements in some more analytical detail.
Specifically, the conclusions about the role of sea-ice albedo and clouds could be supported with some more evidence. From the analysis of sea ice distribution, surface albedo and absorbed short-wave radiation, the authors conclude that changes in cloud properties and ice albedo are responsible for the observed changes in short-wave absorption, but these properties are not shown directly. It would be ideal if the authors could somehow quantify the individual contributions of changes in sea-ice albedo and cloud coverage in order to support their statements. For example, how has the downwelling short-wave flux changed? As several cloud properties are also included in APP-x (for example cloud radiative forcings), it might not be out of proportion for this brief communication to test whether changes in these properties are in line with the author's claims that are based on some other variables of the same dataset.

We thank you for your suggestions in which directions the research could be extended. However, doing so is beyond the scope of this Brief Communication, thus we did not add further analyses. We will pursue these excellent suggestions in the future.

In the conclusions, the authors repeat their statement that the short-wave absorption is increasing. I would also expect some statements about the scientific relevance of this finding, e.g. implications about our understanding of the polar climate, and how future research should address the questions that have to remain open.

We thank you for this suggestion. We added some text as suggested.

Specific comments
- p. 2, l. 14: It is stated here and also at other places (e.g. p. 4, l. 13/14) that the dataset "takes into account" changes in cloud cover, albedo changes, etc. This seems to relate to the statement that "the energy balance is kept closed" (p. 2, l. 22/23) which seems to involve some kind of model. I suggest to explain this in more detail. Why is it an advantage to use this approach instead of independent observations of different variables? For readers not familiar with the available data, it could be explained in more detail which properties are observed and which are inferred by the authors involving additional assumptions.

The processing is described in the methods section "APP-x contains twice-daily data of many surface, cloud, and radiative properties retrieved at high sun and low sun times … from satellite data using a suite of algorithms and a radiative transfer model (Key, Wang et al. 2016)." This presents the appropriate reference, where retrieval details can be found. We do not think that it would be beneficial to include more detailed information about what exact models are used into our text, as this is not central to our conclusions. Nevertheless, we agree that the statement may be unclear, so it has been revised.

- p. 3, l. 8: "accounts for true cloud cover". How? And what is "true"?

We removed the ambiguous word "true".

- Fig. 1: Why is the short-wave absorption much more variable in the Southern hemisphere? Given these fluctuations, is the trend significant at all? If not, it should not be claimed that the solar absorption has been increasing, and the conclusions might be better based on the correlations between variables instead of the trends.

The Southern Hemisphere absorption is much more variable, because it is very sensitive to the large annual cycle of sea ice cover in the southern ocean. Significance of the trend is discussed in the text, while the main important point is that there is no significant opposed trend.

- Fig. 2: I am puzzled by the fact that the sign of the trends is spatially so uniform, even in places where there is no sea ice. How can this be the case? Could there be a bias?

Shortwave absorption trends outside the sea ice zone are caused by changes in cloud cover and atmospheric composition, which are more uniformly distributed than sea-ice. Of course, satellite products can contain biases, however due to the validation of the APP-X albedo and shortwave fluxes we do not think this is the case here.

Minor comments
Main text

- What is new about the APP-x dataset, i.e. what makes it a "novel tool"? Hasn't AVHRR data been available for a long time?

This is described in p2. L.13 "It provides surface radiative properties and fluxes consistently derived twice daily (high and low sun) for both polar regions beginning in 1982". You are absolutely correct that AVHRR has been around for a long time. However it is new in APP-x, that the derived variables are based on methods developed specifically for the polar regions.

- p. 2, l. 12: remove "large"

removed

- p. 3, l. 11: "slightly larger numbers" – how much larger approximately?

Details of the comparison to passive Microwave data were included in our replies to the last review round: http://www.the-cryosphere-discuss.net/tc-2016-279/tc-2016-279-AC3-supplement.pdf

- p. 4, l. 3: rephrase to make clearer: "seasonal Arctic sea ice..., the dataset shows a decrease..., while in the Antarctic, albedo trends show regional differences ... ice concentration."
Rephrased accordingly

- p. 4, l. 11: remove "relatively"
Removed

- p. 4, l. 15: "The mean annual shortwave energy flux...."
Edited accordingly

- p. 4, l. 18/19: Why is the absorption more spatially uniform in the Arctic?
Added "due to uniform sea ice changes"

- p. 4, l. 24: I suggest to cite the source of this result, or explain how this test was done.
A trend, where the 95% confidence interval encompasses zero is not significant at 95% confidence level.

- p. 4, l. 26/27: "energy absorption" – isn't it only the short-wave absorption and not the full energy balance what the authors address here?
Changed to "shortwave energy absorption"

- p. 4, l. 30: "does not show a pattern" – I don't understand what is meant here.
We reworded for clarity

- p. 5, l. 11: "Increases in Antarctic sea ice only occur during the Southern Hemisphere winter". Fig. 1a seems to contradict this statement. Anyways, I do not consider this aspect important enough for the conclusions section. Instead, I would prefer to see a discussion of the implications of the main results (see above).

We added the word "Significant increases…" for clarification. We also added some of your suggestions to the conclusion.

Fig. 1

- Fig. 1: The caption refers to minimal and maximal seasonal extent, but within Fig. 1a it is referred to March and September (by the way, months are written with a capital letter).

This is the correct labelling, as minimal extent occurs in different months in both hemispheres.

Fig. 2

- Caption: trends in "sea ice albedo". Is this really what the figure is showing? As far as I understand it shows the surface albedo change.

Corrected accordingly

- I suggest to use a different colour bar that gives less weight to very small changes and more weight to large changes.

Thank you for the suggestion, the figure is now clearer.

- a) and b) "Arctic" and "Antarctic" are unnecessary details in the variable name because one can see what region it is on the map anyways.

Corrected accordingly

Fig. 3

- This figure is not discussed in the text. I suggest to embed it in the line of argument more explicitly.

The figure is discussed in p.4,ll. 30-34

- "albedo anomaly". Surface albedo, I guess?!

Corrected accordingly

[revised manuscript text omitted]

---

## Author Response (AR3)

**Replies to Editor Comments, 1 August 2017**

We thank you again for your efforts in evaluating our work, for your positive assessment of the latest revision, and for your helpful suggestions. Please find our replies in blue font under your comments.

**Editor's comments:**

1. Last sentence of abstract: This sentence is not very clear to me, and links in particular only weakly to the first sentence. The first sentence only speaks about the Southern Ocean, but the second sentence introduces some generically retreating sea ice. I find this confusing. Maybe the following could work, but please feel free to adjust this as needed: "The observations hence show that the small increase in Antarctic sea-ice extent can not compensate for the combined effect of retreating Arctic sea ice and changes in cloud cover, which both result in a total increase in solar shortwave energy deposited into the polar oceans." Note that I also took out the "surface albedo" in this suggestion, as I found this to be too unspecific to be easily understood here.

Good suggestion, which we adopted only a minor modification.

2. p.3, l.30: Wouldn't it make more sense to turn this around and say "Thus, the slight gains in Antarctic sea-ice area does not compensate for the areal loss of sea ice in the Arctic."

Another excellent suggestion.

3, p.4, l.27 and l.29: Please double check that the error in the analysis indeed causes a difference of exactly a factor 1e5 in the Arctic, but a factor of 1.35e5 in the Southern Ocean. Depending on the type of error, this factor should possibly be the same.

This is correct, the error was caused by some calculation mistake where the result was multiplied with the number of ice covered gridcells. This was incorrect, but as the ice areas are different in both hemispheres, this leads to different ratios between false and corrected numbers.

4. p.4, l.3: I find this sentence not clear. What does it mean that the "Energy flux anomaly shows a relation to interannual variability"? Do you refer to the interannual variability of a specific climate component?

(This occurs on p.5.) We agree that the sentence was not terribly clear. It refers to the patterns in Figure 3. The sentence has been rewritten as "In the Antarctic, the energy flux anomaly does not exhibit a strong relationship with ice extent anomalies overall (Figure 3b,d), while in the Arctic anomalies in albedo and the resulting heat input into the ice-ocean system are much more closely related to the sea ice extent anomaly (Figure 3a,c,e)."

In addition to these comments, I also stand by my impression that the Eisenman et al. ,2014 reference refers to a dataset that is no longer distributed and no longer used by most scientists, and only considers the Antarctic. I am hence still not convinced that it should be used to substantiate the

claim that there is no long-term consistent observational dataset for both poles. I leave it to you to decide on the possible inclusion of this reference.

Agreed. The reference has been moved to an earlier sentence about the increase in Antarctic sea ice extent, with an additional statement "
[revised manuscript text omitted]